# Alloparenting by Helpers in Striped Hyena (*Hyaena hyaena*)

**DOI:** 10.3390/ani13121914

**Published:** 2023-06-08

**Authors:** Ezra Hadad, Amir Balaban, Reuven Yosef

**Affiliations:** 1Israel Nature and Parks Authority, 3 Am Ve’Olamo Street, Jerusalem 95463, Israel; ezra.hadad9@gmail.com; 2Society for the Protection of Nature in Israel, Tel Aviv 66186, Israel; balaban8@gmail.com; 3Eilat Campus, Ben Gurion University of the Negev, Eilat 88100, Israel

**Keywords:** alloparenting, cooperative breeding, helpers, striped hyena, *Hyaena hyaena*

## Abstract

**Simple Summary:**

Helpers are important to help breeding females raise their young in secure and safe surroundings. The helpers are usually young from a previous litter/brood, i.e., the siblings of the cubs being reared. However, grandmothers who are past the breeding stage also help their daughters raise their grandchildren. Here, we present the first ever observations in the wild of helpers in context of the striped hyena. The helpers play, feed, and protect the cubs at the maternal den. The cubs play with the helpers and learn social and other skills which will help them survive as adults.

**Abstract:**

In an ongoing study of the striped hyena (*Hyaena hyaena*), we observed that in the nine different females, alloparenting by the daughters of a previous litter was not uncommon and occurred on fifteen different occasions, twice with two helpers. Alloparenting persisted from when the cubs are approximately a month old until they reach the age when they go out foraging with their mothers at 10–12 months. Helpers perform most maternal duties, except suckling, even in the mother’s presence. Helpers accrued indirect fitness and practiced parenting before reaching sexual maturity. Future studies must study the reproductive biology of the striped hyena in the wild throughout its geographic range to elucidate additional breeding properties that have not yet been identified. The continued persecution of striped hyenas and the lack of information about their breeding rituals and capabilities in the wild mean that this study of their different reproduction strategies, focusing on surrogate mothers, is of great conservation importance. The fact that we have found cooperative breeding in this solitary species suggests that there is much more to uncover of the enigmatic striped hyena in the wild.

## 1. Introduction

Offspring that help their parents raise subsequent clutches/broods is a well-known phenomenon mostly in vertebrates such as fish [1], avian [2,3,4], and mammalian species [5], including humans [6]. These individuals are called “helpers” and the system is termed “cooperative breeding” or “alloparenting”. Helping by non-breeding individuals, wherein young from previous broods remain with the parent(s) and participate in parental duties, is considered altruistic [7,8]. Helpers were usually those individuals that grew slower than their siblings’ growth, were smaller, and more prone to mortality [1]. Hence, it was in their interest to remain in a sheltered and known environment defended by the parents or other members of the group, and in return, to help with the parental duties, which include brood care, provisioning food, teaching survival skills through play, etc. It has also been assumed that because males are responsible for population connectivity and gene flow through dispersal, females are usually those that are helpers, show stronger philopatry, and strengthen local populations [9,10]. Parents that accept helpers appear to have higher rates of annual survival and the authors of [4] concluded that one of the secrets to a long life was to minimize the efforts invested in parental investment.

In mammals, only in ca. 1% of the species have helpers been reported. Initially, it was suggested that alloparenting was restricted to monogamous species [11], but later, the authors contended that it mostly occurred in habitats with low annual rainfall of >50 mm/year, i.e., in arid environments [12]. Further, the authors found that cooperative breeding systems, i.e., with helpers, occurs primarily in rodents, carnivores, and primates; and has evolved where females are socially monogamous, and are polytocous (multiple offspring).

Amongst the cooperatively breeding carnivores, foraging at great distances from the den with relatively sessile young carries the risk of increased predation. Hence, the importance of adequately guarding the young while most of the pack are out foraging becomes important for attaining successful reproduction [13]. Additionally, increased predation rates on the young in smaller groups resulted in markedly reduced fitness levels than in the larger packs and also granted load-lightening to the parents (wild dogs *Lycaon pictus* [14]; gray wolves *Canis lupus* [13]). Load-lightening has been shown to not only increase the survival and growth of the young but also that of the female breeders who were in better body condition [15,16]. Helpers in Meerkats (*Suricata suricata*) also positively influence the survival of older individuals, i.e., those >4 years old who display aspects of senescence [17]. Additionally, the inclusive fitness benefits by helping kin were demonstrated in killer whales (*Orcinus orca)*, wherein post-reproductive grandmothers were shown to improve the survival of their grand-offspring [18]. Similarly, it was found that displaced dominant, senescent females in the Seychelles Warbler (*Acrocephalus sechellensis*) helped to raise group offspring [19].

One of the mechanisms of suppressing subordinate reproductive females in a cooperatively breeding species is infanticide [20]. This favors HPG (Hypothalmic–Pituary–gonadal) conditioning in such females that reduce their fertility to avoid the cost of producing doomed progeny. Thus, it is supposed to favor their investing in helping to raise the dominant females’ young and to gain parental experience until they can branch out and establish their own family groups, or when they eventually become the dominant female. Differences in hormonal levels between dominant and subordinate females have also been demonstrated [21]. The breeding females have higher levels of estrogen, luteinizing hormone, progesterone, and prolactin. They also found elevated levels of prolactin and oxytocin in alloparenting individuals. However, the endocrine studies are limited to only five species, with a focus on Meerkats, and the subject needs to be studied in other cooperatively breeding species [20].

The family Hyaenidae, with only four extant species, is the smallest family in the order Carnivora [22]. The striped hyena and the brown hyena (*Parahyaena brunnea*) are the closest related phylogenetically. Alloparenting has been reported in the latter, wherein females may nurse each other’s young but are known to prioritize their own young in the den, and genetic relatedness determines the degree of cooperation provided [22,23,24]. In contrast, lactating striped hyenas are solely responsible for the care of the litter [22]. However, in Kenya, two sub-adult, non-breeding females were observed frequently at a den site with two newborn cubs and the mother; subsequent genotypic analysis confirmed that the helpers were full-sibling sisters of the younger cubs and daughters of the mother from a previous brood [25]. This mention in a Ph.D. dissertation is the only reference to the possibility of helpers/alloparenting in striped hyenas, but it was not subsequently published in the scientific literature [25]. Further, there are no known studies of breeding in striped hyenas from the wild and the information in the *Handbook of Mammals of the World* is mostly from captive individuals [22]. Here, we present conclusive evidence for alloparenting by striped hyena (*Hyaena hyaena*) helpers in Israel.

Based on our field observations, we questioned whether alloparenting occurred and was a common feature in what was, to date, considered a solitary carnivore, the striped hyena, and hypothesized that we would identify, within a familial group, alloparenting in striped hyenas in Israel as was reported in Africa by Wagner [23,25].

## 2. Materials and Methods

In our ongoing study of striped hyenas [26,27,28] in the Judea region and the lower Judean hills between 1993–1995 and 2006–2023 (Table 1), we have observed and documented with still photographs, camera traps, and video camera thousands of hours of footage and live video recordings documenting parental behavior in the dens of striped hyena.

All dens were located while driving in a 4 × 4 vehicle and following adults or by tracking them on foot. We identified active dens by fresh tracks and food remains. To verify occupancy and confirm breeding, we placed trail cameras at such dens. We conducted observations with a pair of Swarovski 42 × 10 binoculars, 20 × 60 Kowa telescope, and a DSLR video recorder with a 1200 mm telephoto lens.

Our study includes dens documented across central Israel from the lower Judean hills, the Judea region, and the Northern Negev Desert (Figure 1). The climate varies significantly—the coastal regions have cool and rainy winters with long and hot summers; the northern Negev desert region has a semiarid climate with cold winters and hot summers and little rain [29]. Out of conservation concerns, we do not include the names of the dens or the coordinates.

We consider a den as the location where a mother striped hyena gives birth to her litter and raises them for the first few months. Throughout, we use “female” as the breeding mother and “helper” as the surrogate mother, which, in our case, was always a female from a previous litter. We describe a litter as the number of siblings in the same cohort that are present in the den together.

All tests are two-tailed, we present means ± SD and statistical significance was set at *p* < 0.05.

## 3. Results

We successfully documented helpers at the den in 15 (20.3%) of the 74 litters studied. The number of cubs ranged from 1–4. In 13 (86.7%) dens, there was one helper, and at 2 (13.3%) separate dens, there were two helpers each. There was no correlation between the number of cubs in a den and the number of helpers (r^2^ = 0.042).

Because of our ongoing, multiple-year study of the dens of the hyenas in our study area, we are able to identify the individuals based on their body patterns [30], and with certainty, to establish the frequency of alloparenting at the 15 dens. In five (33.3%) of them, helpers were observed only once; in three (20.0%) dens, helpers were observed twice; and one (6.67%) den had helpers in four different years (Table 2). All dens with helpers in multiple years were at the same site and with the same female. Females who had one helper had, on average, fewer cubs (2.1 ± 0.74) compared to those who had two helpers and whose litters were bigger (3.5 ± 0.58), and this was statistically significant (t = −3.609, df 12, *p* < 0.05). Further, females that had multiple instances of helpers had more cubs (3.13 ± 0.64) than females that only had a helper once (1.67 ± 0.52), and this was statistically significant (t = −4.9114, df 12, *p* < 0.05).

A comparison of the litter sizes between females who had helpers (*n* = 15) and those who did not (*n* = 54) shows that with helpers, 53.3% of the females had 3/4 cubs, compared to the 37.0% without helpers (Table 3). Nevertheless, we found no significant differences (Mann–Whitney U test; U = 330.5, *p* = 0.255).

By watching the videos, we were able to generalize several of the cognitive phenotypic behaviors [31] and patterns/rituals that were common to the different dens, as follows:No males participate in the raising of the young or bring food to the den.It is the female’s sole responsibility to raise, feed, and care for the young.The helpers were all female cubs of the mother from a previous litter, i.e., they are full sisters of the cubs they care for and are at least 10–12 months old (Figure 2).Helpers are present with the cubs that are 1 to 10 or 11 months of age (Figure 3), after which, the cubs accompany their mother in her forays for food.Mothers may suckle their young up to the age of 10 months.In several cases, we observed the mother give food she brought to the den to the helper to feed the cubs and herself.After the cubs reached two or three months of age, the mothers slept in a separate den, at distances of 100–500 m, while the helper(s) slept with the cubs.In most dens, the helpers regulate the feeding hours of the mother. At a specific time in the evening, they leave the den to fetch the mother to the cubs.Upon the arrival of the mother at the den, the cubs excitedly run to her and try to suckle. The helper also tries to attract attention by rubbing herself against the mother. However, she does not interfere when the mother lies down to suckle the cubs (Figure 4).When the mother and the helper(s) meet outside the den, they greet each other by sniffing the head and ears and the anal region [32,33]. On some occasions, the helper would lie down, and the mother licked her anal areas (Figure 5).

11.Cubs exit the den when they are ca. a month old and accompanied by the helper if the mother is not present.12.Cubs comb their fur when they are excited or feel threatened.13.They also imitate the behaviors of the helper—we observed a golden jackal (Canis lupaster) approach the den and the helper raised her hackles, the black mane along her back, and all three cubs did the same.14.The mother and the helpers have specific areas where they defecate; some were in the den while others were outside. The cubs quickly learnt to defecate only at these toiletry points.15.During the day, when the mother is away, the helper plays with the cubs and instructs them in survival techniques such as burying food. In areas of loess soil, the helper taught the cubs how to dig and remove earth to enlarge the den. We also saw a helper rubbing her teeth against a bush (to sharpen or to clean them?), and the cubs imitated her actions (Figure 6).

16.The helpers engage the cubs in games with different objects, many times while the mother rested under a bush some distance from the den. The “toys” were mostly of anthropogenic origins and were old shoes, plastic bottles, soft aluminum drink cans, pieces of plastic pipe, etc. However, they also play with bones, skulls, and pieces of skin from food remains brought to the den by the mother. Games usually included running in circles with the cubs chasing the helper and accompanied with light bites, fur/hair pulling, barking or growling, and baring teeth (Figure 7).17.When the mother is able to bring large quantities of food, she will cache the excess under a bush or dig a hole in the ground up to 400 m from the den. During the day, helpers extricate remains from these caches to feed the cubs.

## 4. Discussion

Our field data of breeding female hyenas in the wild showed that litter size ranges from 1–4 cubs and is similar to that mentioned in the *Handbook of Mammals of the World* [22]. We also found that females who had two helpers, and did so over several years, had significantly larger litters than those who had a helper only once or a single helper. However, there are several factors that must be addressed before this is accepted as the norm in this striped hyena population because we are unaware of other important biological parameters that can influence breeding success, such as age. For example, female wolverines (*Gulo gulo*) showed an age-related pattern in reproductive output, with an initial increase followed by a senescent decline in later years [34]. In our case, the lack of the age of the breeding females prevents us from deducing whether the studied individuals were also influenced by age. Additionally, variations in the environment and population density are known to influence reproductive output [35]. Populations of striped hyenas in Israel have shown a marked increase along with exploitation of human resources [26,27,28], which could also influence reproductive output and breeding strategies.

We can deduce from our study that the helpers are helping their younger siblings to survive; to learn crucial survival tasks and techniques through play and imitation; with food provisioning while the mother is resting or foraging; with den maintenance; and to guard them from potential predators. The importance of the environment, including the interactions the young experience in their developmental stages, is recognized since Darwin [36] in the context of phenotypic plasticity [37]. Developmental phenotypic plasticity allows individuals to overcome environmental challenges and is the result of interplay between the organisms’ genetics and the environment it experiences during development [38]. Hence, the contribution of the helpers to their mother considerably enhances the chances of survival of the cubs after their dispersal from the natal areas [17] and improves the mothers’ body condition, which is important for subsequent litters [15,16]. This substantiates the kin selection theory, wherein the inclusive fitness of the helper(s) is increased by helping their mother raise future siblings [39,40,41]; likewise, grandmothers help their daughters raise the grandchildren [4,19].

Many studies of cooperative breeding have found that the presence of helpers results in greater reproductive success [15,16,17]. However, Woodroffe and Macdonald [42] contradict these findings and contend that these studies have not demonstrated a causal relationship. In their study of cooperatively breeding European badgers (*Meles meles*), they found that the relationship between the number of helpers and reproductive success was spurious. They contended that it is a function of territory quality and that helpers appear to have a negative effect on the groups’ overall reproductive success. Our study does not support this contention; we found that the adult females who had a number of helpers and held sequentially over several years had better reproductive success than those without helpers.

Additionally, the alloparenting helpers accrue indirect fitness benefits [43] and gain shelter, food, and experience at parenting while remaining in their natal area and group [39,40,44]. Unlike Owens and Owens [23], who observed alloparenting in the communal Brown hyenas wherein adult females preferentially helped their kin within the clan, we have found that only daughters from an earlier litter, and who are almost a year old, help their mothers in the case of the striped hyena. This is probably very important to the helpers because sub-adult females disperse at far greater distances than males in order to prevent inbreeding and competition between relatives for resources [45,46]. Moreover, striped hyenas are reported to attain sexual maturity in their second year [22], allowing the sub-adult females to act as surrogate mothers and to enrich their parental experience before they disperse and breed.

We also take exception to the findings that alloparenting would occur mostly in areas with low annual rainfall of >50 mm/year, i.e., in arid environments [12]. Our study does not substantiate these findings because the average annual rainfall in the Judea Region is 550 mm/year (https://ims.gov.il/he/AccumulatedRain, accessed on 20 March 2023), and it is 350 mm/year in the Northern Negev [47].

Additionally, several recent studies have demonstrated different degrees of social behavior in the striped hyena, including parental care by adult males [25,32,48]. In all our footage and observations, we have never observed social gatherings near the dens, male attendance in any manner, or exceptional attention by another adult hyena at the den [33]. Our observations confirm that during the breeding period, the adult female at the den is solitary, elusive, and not exclusively nocturnal. This also raises the question of differences in habitats, resources, and other life cycle parameters to reduce inter- or intra-specific competition [49] to maximize fitness. However, we are unable to determine the activities of the females when away from the den daily for several hours, potentially socializing with other adults at a distance; this remains to be studied.

The fact that no other studies have definitively reported—except for the one unpublished observation [25]—the relatively high occurrence of alloparenting by helpers in striped hyenas may be because it is, in general, a rare phenomenon across the species range. However, one must also take into account that the number of studies on breeding in striped hyenas is negligible to non-existent; most studies address the ecology—that is, food/prey [48,49,50], den locations [51], physiology [52], distribution and densities [53,54], threats and conservation [55,56], social interactions [45,57], population trends [26], human-hyena conflict [58], etc.

## 5. Conclusions

We observed that in nine different striped hyena adult females, alloparenting by their daughters of a previous litter was a relatively common phenomenon, occurring on fifteen different occasions, twice with two helpers. The helpers alloparent from when the cubs are approximately a month old until they reach the age when they go out foraging with their mothers, i.e., 10–12 months. Helpers do most of the maternal duties—except suckling—even in the presence of the mother. Helpers accrue not only indirect fitness but also practice parenting before reaching sexual maturity, which occurs at two years of age. Future studies must understand the reproductive biology of the striped hyena in the wild throughout its geographic range to elucidate additional breeding capabilities and strategies which have not yet been identified. The continued persecution of striped hyenas globally [59,60] and the fact that their breeding rituals and capabilities in the wild remain unstudied [61] mean that the study of their reproduction is of great conservation importance. The fact that we have found cooperative breeding in this solitary species suggests that there is a lot more to uncover of this enigmatic species in the wild.

## Figures and Tables

**Figure 1 animals-13-01914-f001:**
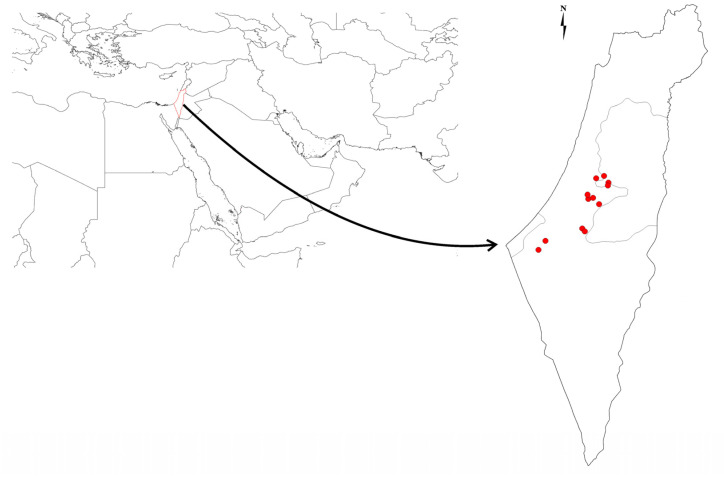
The locations of the studied dens of striped hyena (*Hyaena hyaena*) in Israel.

**Figure 2 animals-13-01914-f002:**
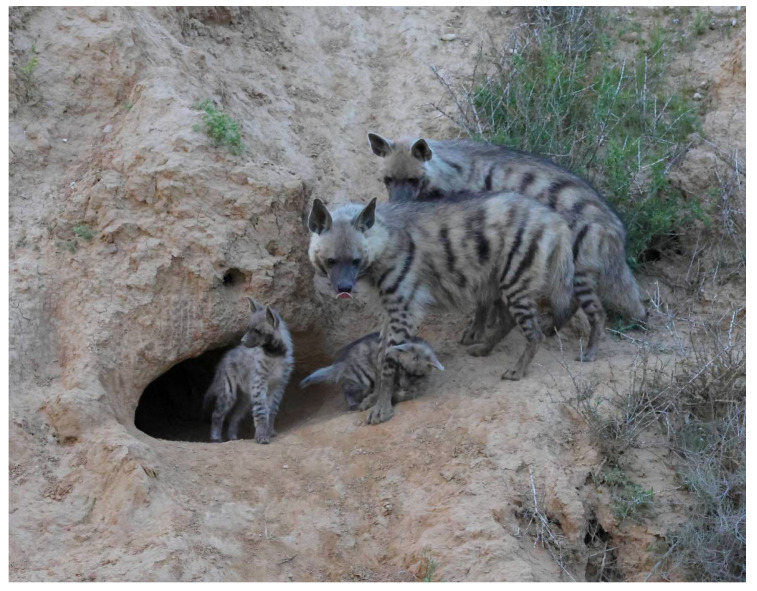
Mother in front, the helper looking on, and two striped hyena (*Hyaena hyaena*) cubs at the entrance to their den. Photo: Ezra Hadad.

**Figure 3 animals-13-01914-f003:**
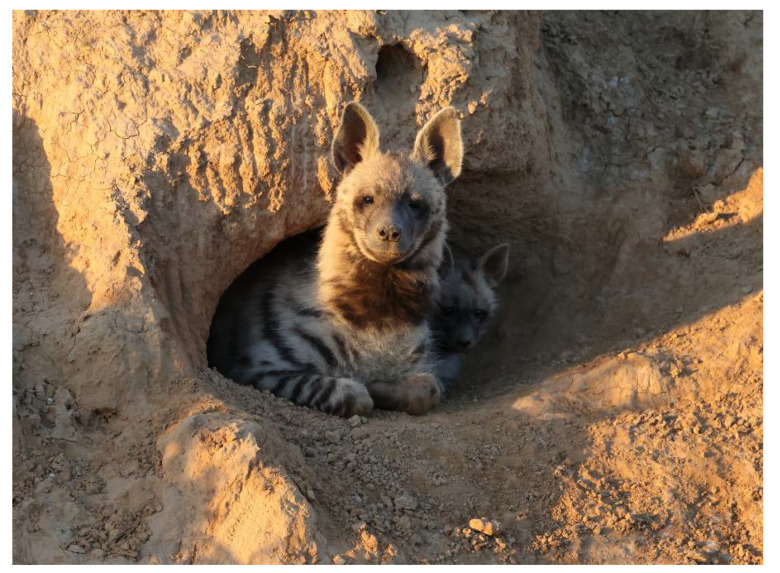
A helper striped hyena (*Hyaena hyaena*) at the entrance to the den with a cub. Photo: Ezra Hadad.

**Figure 4 animals-13-01914-f004:**
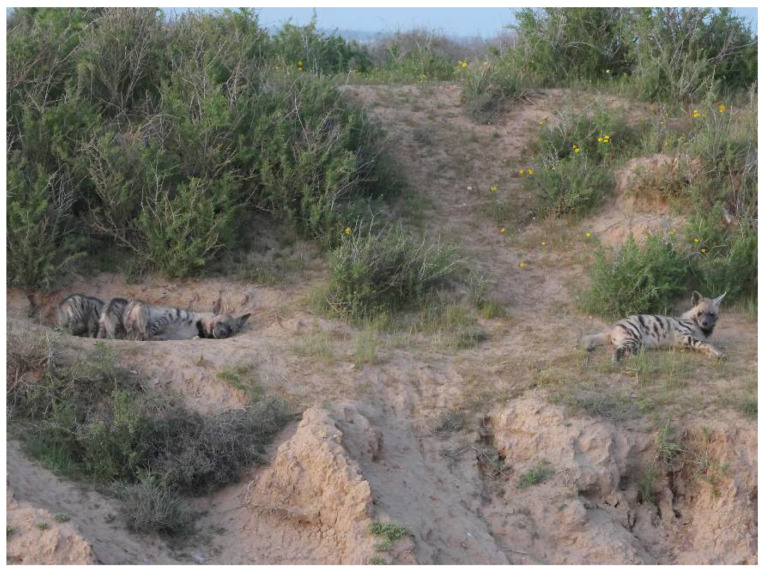
A female striped hyena (*Hyaena hyaena*) suckles her three young while the helper rests at a distance. Photo: Ezra Hadad.

**Figure 5 animals-13-01914-f005:**
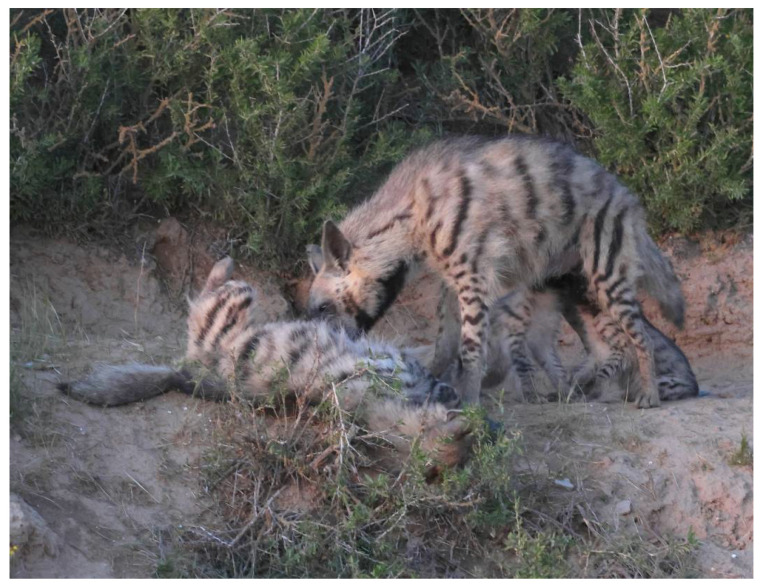
The helper prostrating herself and being licked in the anal region by the mother striped hyena (*Hyaena hyaena*) upon her arrival. Note the two cubs suckling from the mother. Photo: Ezra Hadad.

**Figure 6 animals-13-01914-f006:**
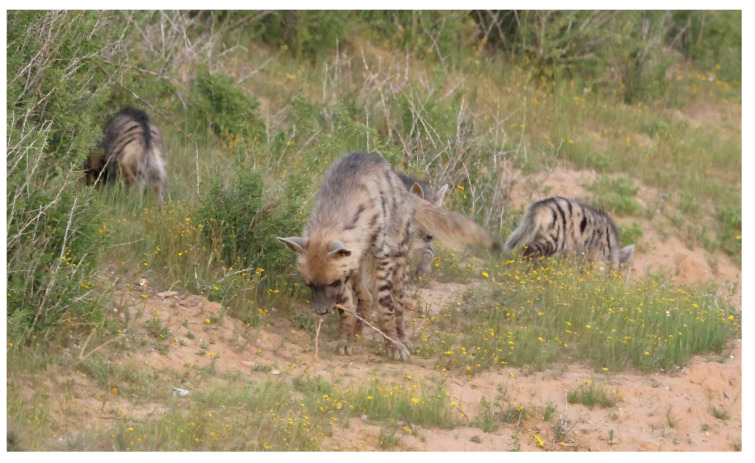
The helper striped hyena (*Hyaena hyaena*) biting on a twig. Photo: Ezra Hadad.

**Figure 7 animals-13-01914-f007:**
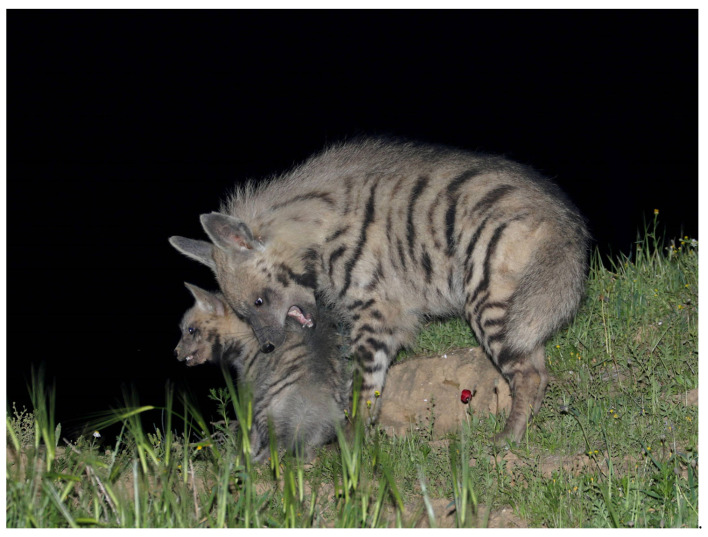
The helper at play with a striped hyena (*Hyaena hyaena*) cub. Photo: Ezra Hadad.

**Table 1 animals-13-01914-t001:** Number of striped hyena (*Hyaena hyaena*) dens observed in the study years, 1993–1995 and 2006–2023 in Israel. w/o denotes without helpers; w/is with helpers.

Year	w/o Helpers	w/Helpers
1993	2	
1994	3	
1995	1	
2006		1
2007	1	
2008	1	
2009	2	
2010	2	
2011	3	1
2014	2	
2015	3	2
2016	3	
2017	2	1
2018	6	
2019	6	2
2020	1	3
2021	3	1
2022	10	1
2023	8	3
Total	59	15

**Table 2 animals-13-01914-t002:** A comparison of females with one helper vs. those with two helpers. The number in the female column identifies those who had helpers in multiple years.

Female	Year	No. Helpers	Litter Size
1	2011	1	1
2	2015	1	1
3	2015	1	2
4	2017	1	2
4	2019	1	3
4	2020	1	3
4	2021	1	3
5	2019	2	3
5	2020	2	4
6	2021	2	3
6	2023	2	4
7	2022	1	2
8	2023	1	2
9	2023	1	2

**Table 3 animals-13-01914-t003:** Comparison of litter sizes between female striped hyena (*Hyaena hyaena*) with and without helpers.

Litter Size	w/Helpers	%	w/o Helpers	%
1	2	13.3	11	20.4
2	5	33.3	23	42.6
3	6	40.0	16	29.6
4	2	13.3	4	7.4
*n*	15		54	
Average	2.53		2.24	
S.D.	0.916		0.867	

## Data Availability

All data are included in the paper. Exact locations of dens are not disclosed out of conservation concern and ongoing persecution of the study species.

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
