# Peer review of "Alloparenting by Helpers in Striped Hyena (Hyaena hyaena)"

_animals, 2023, doi:10.3390/ani13121914_

Round 1
Reviewer 1 Report
This is an interesting manuscript documenting, related subadult helpers in the striped hyena. The fact that helpers have never been reported in this species is surprising since there are many published papers and species reviews.
My main comments on this manuscript are in organization and information presented in the introduction, the lack of specificity ion the methods, the lack of clarity in the results and the free-flowing discussion (which is not sub-headed as far was I could see). The paper needs a bit of revision to insure that it has the impact the authors intend - the report of a new behavior in a species of conservation concern. There are also numerous grammatical issues that need to be addressed.
General Comments: Introduction - This is generally good but very broad. I'd suggest that a paragraph about striped hyena reproductive biology would assist the non-hyena specialist in understand the issue. For example, my understanding is that striped hyena are socially, if not, genetically monogamous and that the male actually stays with the female through birth of the cubs and even helps with the rearing. Additionally, the what is the sexual maturity age of young hyenas and when do young hyenas disperse. Also what is the sex ratio of litters. I think the you need to just have a brief summary so the reader has some context.
Methods - This section should really be expanded. I realize you do not want to identify precise den locations (I assume they are used multiple years?) but you indicate that your study included wetter coastal areas to more semi arid areas in methods, but later only mention, what I assume are, wetter areas? I really think a general map of the general study area is critical. Somewhere you mention that there are 550mm of rain in you area, but I doubt that is in the Negev. You write that your study went from 2006-2023 (17-18 years) but only report helpers over a 13 year span (2011-2023) and only observed helpers in 8 years (less than 50% of the time?). I think more thought and precision can help.
Results - This section cries out for more data reporting. You need to be clear in what you mean by females, dens, litters. My reading is that there were nine adult females that had at least one helper, in 15 dens (litters - assuming that one den= one litter). You should consider a new table that does not just include litter size but also includes more information: Maybe something like:
Female Year. Helpers. Litter size
How many females did you actually have in your study? You do report that you had nine with helpers and there were 15 dens, which to me means that some females had helpers in multiple years. Please present this data in a clear manner that a reader can follow. Are some females more open to having helpers? When do the adult males leave the females at the den? There is so much more information you must have that is not reported that may help explain your results.
Discussion - I assume that it begins on line 205? Why not include a subheading? You have lots of interesting information in general here but little follows what you present in the results. The discussion is best if it discusses you results in order of presentation. I don't doubt that you document helpers, just that you really do not have any conclusive idea of why they occur, even based on other studies.
Specific comments:
Line 23-25: Very awkward sentence construction - rewrite.
Line 26: add "that there is"
Line 33: add "is" after system
Liner 34-35: We can debate this, but is it really altruistic behavior if the helpers "gain" from helping? Also, in your case, unless I have this incorrectly, the subadults are not repoductive at 11-12 months of age so they are not putting off their own reproduction.
Line 75: delete one of the "higher levels"
Lines 80-82: Placement of this sentence. I'm a bit unsure of its significance, but if you want to keep it I'd move it earlier in the section. Delete it or move it.
Line 84: You are writing about taxonomy here so I'd write something like, "The family Hyaenidae, with only four extant species is the smallest family in the order Carnivora."
Line 87: "...but "are" known to prioritize..." should be are not is.
Line 88-89: "In contrast "lactating" striped hyena..." I'd add lactating.
Line 90: Are subadults reproductive? You need to clarify this.
Line 93: "This mention in "a" dissertation..." I'd add a. Also please site the dissertation here.
Line 98: I'd delete "Hence" - it does not add to the sentence.
Line 99: I'd write, "occurred and was common..."
Line 98-100: Very awkward construction.
Line 101-102: The is not a scientific hypothesis. I'd suggest rewriting this in a hypothetic-deductive manner of change the wording. For example, if it really is your hypothesis that you would find familial alloparenting - please add why you think you should find this.
Line 109: add "placed" not place.
Line 116: Do you mean "not"? You left it out. Also, I really think a general map (not showing den sites would help.
Line 121: Much of this is redundant to what you write in methods.
Line 122-125: Be clear on dens vs. litters vs. females.
Line 128-130: The first dssentence seems like something that should be in methods.
Line 130-133: You are really writing about females (9). I'm interested in females having multiple instances of helpers vs. females that only may have a helper once.
Line 137: I'd delete this last part about small sample size.
Line 146: Why not delete "invariable" and just write, "The helpers were all females..."
Line 147: Photo 1 - Unless you are a hyena specialist there is no way to know which large hyena is the adult. Can you add something here?
Line 151: I love the photo, but do not know what this photo has to do with what you write.
Line 156: Do you need "sometimes"
Line 157: Delete "the"
Line 159: Data on this? Did it occur in all dens in all years?
Comment: I don't dislike the generalizations, but not sure where they come from. After 18 years of study (really 8 years of helper observations) you must have data that could support these generalizations. Otherwise the generalizations are not supported.
Line 205: did I miss the Discussion? Maybe line 205?
Line 213-218: You must have data on this.
Line 226: Add helpers after alloparenting.
Line 239: Is 15/69 relatively high? Nine of how many females?
Conclusions: I'd delete "prove" maybe use observed?
There are some minor poor choices of grammar and a few mistakes.
Author Response
This is an interesting manuscript documenting, related subadult helpers in the striped hyena. The fact that helpers have never been reported in this species is surprising since there are many published papers and species reviews. – we agree with this sentiment and were surprised ourselves at the lack of breeding data on the species in the wild.
My main comments on this manuscript are in organization and information presented in the introduction, the lack of specificity ion the methods, the lack of clarity in the results and the free-flowing discussion (which is not sub-headed as far was I could see). The paper needs a bit of revision to insure that it has the impact the authors intend - the report of a new behavior in a species of conservation concern. There are also numerous grammatical issues that need to be addressed. – thank you for your help
General Comments: Introduction - This is generally good but very broad. I'd suggest that a paragraph about striped hyena reproductive biology would assist the non-hyena specialist in understand the issue. For example, my understanding is that striped hyena are socially, if not, genetically monogamous and that the male actually stays with the female through birth of the cubs and even helps with the rearing. – in our study, unlike others which we have specified, males do not partake in the raising of the young.
Additionally, the what is the sexual maturity age of young hyenas and when do young hyenas disperse. – young disperse when they are about a year old. because females can breed in their second year, although how this influences the helpers who defer to help their mothers, remains to be studied. Included in discussion.
Also what is the sex ratio of litters. I think the you need to just have a brief summary so the reader has some context. – We do not have an answer to these questions. Once again, as specified in our paper, the sparse knowledge on breeding in the study species is from captive individuals and not from the wild.
Methods - This section should really be expanded. I realize you do not want to identify precise den locations (I assume they are used multiple years?) but you indicate that your study included wetter coastal areas to more semi arid areas in methods, but later only mention, what I assume are, wetter areas? I really think a general map of the general study area is critical. Somewhere you mention that there are 550mm of rain in you area, but I doubt that is in the Negev. – Now included
You write that your study went from 2006-2023 (17-18 years) but only report helpers over a 13 year span (2011-2023) and only observed helpers in 8 years (less than 50% of the time?). I think more thought and precision can help. – we have reworded this section but feel that it is important for the reader to realize that these observations are a subsection of a much larger study of the striped hyenas.
Results - This section cries out for more data reporting. You need to be clear in what you mean by females, dens, litters. My reading is that there were nine adult females that had at least one helper, in 15 dens (litters - assuming that one den= one litter). You should consider a new table that does not just include litter size but also includes more information: Maybe something like:
Female Year. Helpers. Litter size -- New table, information, and analyses included
How many females did you actually have in your study? You do report that you had nine with helpers and there were 15 dens, which to me means that some females had helpers in multiple years. Please present this data in a clear manner that a reader can follow. Are some females more open to having helpers? –Now specified. Table 1.
When do the adult males leave the females at the den? There is so much more information you must have that is not reported that may help explain your results. – males have nothing to do with the rearing of the young as reported in our paper.
Discussion - I assume that it begins on line 205? Why not include a subheading? You have lots of interesting information in general here but little follows what you present in the results. The discussion is best if it discusses you results in order of presentation. I don't doubt that you document helpers, just that you really do not have any conclusive idea of why they occur, even based on other studies. – Section now included and expanded
Specific comments:
Line 23-25: Very awkward sentence construction - rewrite. – Reworded.
Line 26: add "that there is" - included
Line 33: add "is" after system - included
Liner 34-35: We can debate this, but is it really altruistic behavior if the helpers "gain" from helping? Also, in your case, unless I have this incorrectly, the subadults are not repoductive at 11-12 months of age so they are not putting off their own reproduction. – True in our case but documented in others as referenced.
Line 75: delete one of the "higher levels" - deleted
Lines 80-82: Placement of this sentence. I'm a bit unsure of its significance, but if you want to keep it I'd move it earlier in the section. Delete it or move it. – Partially deleted
Line 84: You are writing about taxonomy here so I'd write something like, "The family Hyaenidae, with only four extant species is the smallest family in the order Carnivora." – reworded as suggested
Line 87: "...but "are" known to prioritize..." should be are not is.- CORRECTED
Line 88-89: "In contrast "lactating" striped hyena..." I'd add lactating. - REWORDED
Line 90: Are subadults reproductive? You need to clarify this. – Added non-breeding
Line 93: "This mention in "a" dissertation..." I'd add a. Also please site the dissertation here.- Included
Line 98: I'd delete "Hence" - it does not add to the sentence. - Deleted
Line 99: I'd write, "occurred and was common..." - Included
Line 98-100: Very awkward construction. – Merged the two sentences
Line 101-102: The is not a scientific hypothesis. I'd suggest rewriting this in a hypothetic-deductive manner of change the wording. For example, if it really is your hypothesis that you would find familial alloparenting - please add why you think you should find this. – Now modified
Line 109: add "placed" not place. - added
Line 116: Do you mean "not"? You left it out. Also, I really think a general map (not showing den sites would help. – Thank you! Now included.
Line 121: Much of this is redundant to what you write in methods. – Attempted to clear up some of the repititions
Line 122-125: Be clear on dens vs. litters vs. females. – Now described in a separate paragraph.
Line 128-130: The first dssentence seems like something that should be in methods. - MOVED
Line 130-133: You are really writing about females (9). I'm interested in females having multiple instances of helpers vs. females that only may have a helper once. – Now included
Line 137: I'd delete this last part about small sample size. - Deleted
Line 146: Why not delete "invariable" and just write, "The helpers were all females..." - AMMENDED
Line 147: Photo 1 - Unless you are a hyena specialist there is no way to know which large hyena is the adult. Can you add something here? – Now specified
Line 151: I love the photo, but do not know what this photo has to do with what you write. – Illustrates role of helper when mother is absent
Line 156: Do you need "sometimes" - reworded
Line 157: Delete "the"- Deleted
Line 159: Data on this? Did it occur in all dens in all years? – happens in most dens hence reworded accordingly
Comment: I don't dislike the generalizations, but not sure where they come from. After 18 years of study (really 8 years of helper observations) you must have data that could support these generalizations. Otherwise the generalizations are not supported. – As mentioned in the methods, these come from observing 100s of hours of footage from the various dens and many are difficult to quantify owing to the overwhelming amount of information. We are trying to keep this paper as a natural history observation with a previously unrecorded behavior, except for that of Wagner, as the focus.
Line 205: did I miss the Discussion? Maybe line 205? – THE PREVIOUS HEADING WAS RESULTS & DISCUSSION. NOW SEPARATED
Line 213-218: You must have data on this. – IF YOU MEAN ABOUT FITNESS AFTER DISPERSAL OR THE MOTHERS’ PHYSICAL CONDITION, NO WE DON’T HAVE THE DATA BECAUSE OUR STUDY IS BASED EXCLUSIVELY ON VISUAL OBSERVATIONS WITHOUT HANDLING (MEASURING) ANY OF THE STUDY ANIMALS.
Line 226: Add helpers after alloparenting. – ADDED.
Line 239: Is 15/69 relatively high? Nine of how many females? – Presented differently
Conclusions: I'd delete "prove" maybe use observed? – changed as suggested

Reviewer 2 Report
Thanks to the editorial board to provide me the opportunity to review the paper title “Alloparenting by helpers in Striped Hyena (Hyaena hyaena)”. The submitted manuscript is excellent work compiled by the authors.
Alloparenting in Striped Hyenas is a fascinating behavior where individuals other than the parents assist in rearing the young. It showcases the cooperative nature of these social animals. The presence of alloparenting helpers in Striped Hyena societies highlights the importance of extended family or group dynamics in the successful upbringing of offspring. Alloparenting is a valuable strategy for Striped Hyenas, as it allows parents to distribute the workload of childcare among other members of their social group, ensuring the survival and well-being of the young. The alloparenting helpers in Striped Hyenas often include older siblings, close relatives. This cooperative approach strengthens family bonds and fosters a sense of shared responsibility. By participating in alloparenting, helpers in Striped Hyenas gain valuable experience in cubs cares, which can benefit them when they become parents themselves in the future. Alloparenting also provides learning opportunities for the young hyenas, as they are exposed to different caregivers who can impart valuable skills and knowledge necessary for their future survival. The presence of alloparenting helpers in Striped Hyenas may increase the overall reproductive success of the group, as it allows the parents to invest more time and energy into mating and securing resources, knowing that their offspring are being cared for by trusted individuals. Alloparenting in Striped Hyenas can be seen as a form of cooperative breeding, where the collective effort of the group ensures the survival and success of the next generation. It is interesting to note that the helpers in alloparenting situations often exhibit behaviors such as grooming, playing, and even defending the young hyenas, showing their dedication to the well-being of the group as a whole. Overall, alloparenting by helpers in Striped Hyenas exemplifies the complexity and adaptability of social structures in the animal kingdom. It is a remarkable strategy that underscores the significance of cooperation and shared responsibilities in raising offspring.
I have few observations which can help to improve the manuscript. These questions must be included in the introduction or in discussion.
What are the primary benefits of alloparenting in Striped Hyenas for both the parents and the helpers?
How do Striped Hyenas determine which individuals will become alloparenting helpers? Is there a specific selection process or criteria involved?
What are the typical roles and responsibilities of alloparenting helpers in the upbringing of young Striped Hyenas? How do they contribute to the care and development of the offspring?
Are there any variations in the level of involvement among alloparenting helpers in Striped Hyena groups? Do some individuals play a more active role than others, and if so, what factors contribute to these differences?
Do alloparenting helpers in Striped Hyenas exhibit any specialized behaviors or adaptations when caring for the young? How do they communicate and interact with the offspring?
Are there instances where alloparenting helpers in Striped Hyenas take over the complete care of the young, temporarily or permanently? If so, what circumstances lead to such situations?
How does alloparenting affect the social dynamics and relationships within Striped Hyena groups? Does it influence the hierarchical structure or bonding between individuals?
Are there any observed instances of conflict or competition among alloparenting helpers in Striped Hyenas? How do they resolve disputes or challenges related to caring for the young?
Are there any known instances where alloparenting helpers in Striped Hyenas switch their caregiving roles between different litters or individuals? If so, what triggers these changes?
What are the long-term benefits of alloparenting in Striped Hyenas? Does it have any impact on the survival rates or reproductive success of the offspring, or the overall fitness of the group?
Author Response
I have few observations which can help to improve the manuscript. These questions must be included in the introduction or in discussion. – THANK YOU FOR YOUR QUESTIONS AND SUGGESTIONS. WE HAVE INCLUDED SOME IN OUR DISCUSSION FOR FUTURE RESEARCH, AND IN OTHER GEOGRAPHICAL REGIONS WHERE SOCIALIZING HAS BEEN OBSERVED.
What are the primary benefits of alloparenting in Striped Hyenas for both the parents and the helpers? –FITNESS LEVELS WERE GREATER. ALTHOUGH OUR SAMPLE SIZE IS RELATIVELY SMALL, THERE SEEMS TO BE A TREND WHEREIN MOTHERS WITH HELPERS HAD MORE YOUNG (Table 1). WE ALSO SUSPECT FEMALES WITH HELPERS INVESTED LESS ENERGY IN GUARDING, AND PREDATION OR LOSS OF YOUNG TO INCLEMENT WEATHER OR OTHER ENVIRONMENTAL FACTORS WAS NOT OBSERVED. MOST OTHER PARENTAL DUTIES WERE PART OF THE REPERTOIRE OF THE HELPER.
How do Striped Hyenas determine which individuals will become alloparenting helpers? Is there a specific selection process or criteria involved? – WE ARE UNABLE TO ANSWER THIS QUESTION.
What are the typical roles and responsibilities of alloparenting helpers in the upbringing of young Striped Hyenas? How do they contribute to the care and development of the offspring? – THE 17 DIFFERENT ROLES/BEHAVIORS OF THE HELPERS IS LISTED IN THE RESULTS SECTION.
Are there any variations in the level of involvement among alloparenting helpers in Striped Hyena groups? Do some individuals play a more active role than others, and if so, what factors contribute to these differences? –WE DID NOT DISCERN ANY DIFFERENCES. THEY WERE QUITE UNIFORM IN THEIR BABY-SITTING DUTIES AND WERE ESSENTIALY SUROGATE MOTERS IN EVERY POSSIBLE MANNER AND FUNCTION.
Do alloparenting helpers in Striped Hyenas exhibit any specialized behaviors or adaptations when caring for the young? – WE SUSPECT THEY ARE GAINING PARENTAL EXPERIENCE WITH THEIR YOUNGER SIBLINGS PRIOR TO DISPERSING THEMSELVES.
How do they communicate and interact with the offspring? – SAME AS THE MOTHERS.
Are there instances where alloparenting helpers in Striped Hyenas take over the complete care of the young, temporarily or permanently? If so, what circumstances lead to such situations? – NEVER WAS THIS OBSERVED.
How does alloparenting affect the social dynamics and relationships within Striped Hyena groups? Does it influence the hierarchical structure or bonding between individuals? – WE DID NOT OBSERVE ANY GROUPINGS OF HYENAS SUCH THAT THESE QUESTIONS ARE NOT APPLICABLE.
Are there any observed instances of conflict or competition among alloparenting helpers in Striped Hyenas? How do they resolve disputes or challenges related to caring for the young? – WE DID NOT OBSERVE ANY DISPUTES. THE HELPER WAS ALWAYS SUBORDINATE TO HER MOTHER – SEE POINT 10, PICTURE 4.
Are there any known instances where alloparenting helpers in Striped Hyenas switch their caregiving roles between different litters or individuals? If so, what triggers these changes? – NO SUCH INFORMATION EXISTS NOR DOCUMENTED IN OUR STUDY.
What are the long-term benefits of alloparenting in Striped Hyenas? Does it have any impact on the survival rates or reproductive success of the offspring, or the overall fitness of the group? – WE HAVE NO KNOWLEDGE ABOUT SURVIVAL RATES AFTER THE DISPERSAL OF THE YOUNG, INCLUDING THE HELPERS. Once again, no adult group activity was documented in our study and we maintain that breeding in Striped Hyenas is solitary.

Round 2
Reviewer 1 Report
Much improved manuscript. Your improvements make it easier for every reader to understand and increases the significance of your work.
I'm still not convinced that sub-adult helpers at the den are really altruistic, but I suppose we can debate that point. One thing that is still missing is litter size in dens with no helpers or any data on fitness of cubs from dens with helpers and those without helpers. Maybe you have that in a separate paper? Also a map of the general area for non-middle east reader still may help. I realize your study is long-term from 2006, but note that your data only start many years later. Not critical but raises questions. Were there no helpers earlier? Why are there helpers now? What also may help is how many females were observed with cubs each year and how many had helpers vs. how many did not. For example, in 2023 females 6, 8, 9 had litters with helpers but you don't say how many other females you documented with litters. Three out of five litters would be more significant than 3 out of 20. Just a thought.
A few specific comments:
1. Still do not like use of "prove" rather than observed in the abstract. You must like it.
2. Not sure what the ?? are in line 98.
3. line 119 - I'd add "which" before in our study.
4. Table 1 is helpful. However, it reports that female 4 had 2 litters in 2020. Is this correct?
5. I think the big thing you find, aside from helpers in your population, is that some females are more prone to having helpers than others. I hope you follow this up. It will be interesting if 6, 8 and 9 have helpers in 2024.
6. Line 258 - Should you put adult before female? I think in brown hyenas it is other adult mothers that help. And it is a bit confusing since your helpers are "kin" - genetically related - to the new cubs. Reread what you write and see what you think.
7. line 271-272 - This helps, but can be confusing since you indicate that alloparenting in striped hyenas has only been reported once in a dissertation and not published then you write it has been observed (unless you combine it with all social behaviors) by adults in other studies. It is critical to your work that you only observed subadults helping. Maybe just a bit more clarity?
8. The generalizations are fine in a review paper or encyclopedia article of striped hyenas. However, in a research article they are just anecdotal descriptions without data. I love descriptive behavioral ecology and think it has a role in science but it is descriptive and not hard to justify without data. Just a thought.
English is fine. Just a few minor adjustments.
Author Response
Much improved manuscript. Your improvements make it easier for every reader to understand and increases the significance of your work. – Thank you.
I'm still not convinced that sub-adult helpers at the den are really altruistic, but I suppose we can debate that point. – Deleted in discussion where relevant.
One thing that is still missing is litter size in dens with no helpers – This is mentioned in Table 2.
or any data on fitness of cubs from dens with helpers and those without helpers. Maybe you have that in a separate paper? – Not yet. But we mention that in all dens a 100% of the young reached to become sub-adults and to disperse. Their subsequent survival and fitness is unknown.
Also a map of the general area for non-middle east reader still may help. – Now included.
I realize your study is long-term from 2006, but note that your data only start many years later. Not critical but raises questions. Were there no helpers earlier? Why are there helpers now? – The helpers were also documented in - Suspect that unawareness or oversight may have resulted in our not noting the behavior earlier. Our earliest observation was in 2006 and once we realized what was happening were better prepared at subsequent dens. Now included in discussion.
What also may help is how many females were observed with cubs each year and how many had helpers vs. how many did not. For example, in 2023 females 6, 8, 9 had litters with helpers but you don't say how many other females you documented with litters. Three out of five litters would be more significant than 3 out of 20. Just a thought. – New Table 1 included to supply this information
A few specific comments:
- Still do not like use of "prove" rather than observed in the abstract. You must like it. - Changed
- Not sure what the ?? are in line 98. - Deleted
- line 119 - I'd add "which" before in our study. – Added.
- Table 1 is helpful. However, it reports that female 4 had 2 litters in 2020. Is this correct? – Thank you for catching that! Corrected to 2021.
- I think the big thing you find, aside from helpers in your population, is that some females are more prone to having helpers than others. I hope you follow this up. It will be interesting if 6, 8 and 9 have helpers in 2024. – Agree, we hope to widen the study by having additional rangers identify dens in their areas and hopefully in the future we will have more data on the subject.
- Line 258 - Should you put adult before female? I think in brown hyenas it is other adult mothers that help. – Kruuk mentions that these are usually sisters that preferentially feed their own young first.
And it is a bit confusing since your helpers are "kin" - genetically related - to the new cubs. Reread what you write and see what you think. – Added.
- line 271-272 - This helps, but can be confusing since you indicate that alloparenting in striped hyenas has only been reported once in a dissertation and not published then you write it has been observed (unless you combine it with all social behaviors) by adults in other studies. It is critical to your work that you only observed subadults helping. Maybe just a bit more clarity? – Now included at several points in Abstarct & Discussion.
- The generalizations are fine in a review paper or encyclopedia article of striped hyenas. However, in a research article they are just anecdotal descriptions without data. I love descriptive behavioral ecology and think it has a role in science but it is descriptive and not hard to justify without data. Just a thought. – In principle I agree but it is a chore to itemize the 1000s of hours of footage we have and to put numbers to each. Hence, we opted for generalizations of the most commonly observed behaviors.
